| Open Peer Review | Host-Microbial Interactions | Methods and Protocols

# Virtual Colon: spatiotemporal modeling of metabolic interactions in a computational colonic environment

Georgios Marinos,[1,2] Johannes Zimmermann,[1,3,4,5] Jan Taubenheim,[1] Christoph Kaleta[1]

**ABSTRACT**  Host-microbial metabolic interactions have been recognized as an essential factor in host health and disease. Genome-scale metabolic modeling approaches have made important contributions to our understanding of the interactions in such communities. One particular such modeling approach is BacArena, in which metabolic models grow, reproduce, and interact as independent agents in a spatiotemporal metabolic environment. Here, we present a modeling application of BacArena, a virtual colonic environment, which reveals spatiotemporal metabolic interactions in a computational colonic environment. This environment resembles the crypt space together with the mucus layers, the lumen, and fluid dynamics. Our proof-of-principle experiments include mono-colonization simulations of context-specific colonic cells and simulations of context-specific colonic cells with the SIHUMIx minimal model microbiome. Our simulations propose host-microbial and microbial-microbial interactions that can be verified based on the literature. Most importantly, the Virtual Colon offers visualization of interactions through time and space, adding another dimension to the genome-scale metabolic modeling approaches. Lastly, like BacArena, it is freely available and can be easily adapted to model other spatially structured environments (http://www.github.com/maringos/VirtualColon).

**IMPORTANCE** Interactions between the human body and gut microbes are crucial for health and disease. We present the Virtual Colon, an extension of the individual-based microbiome modeling approach BacArena that mimics key features of the colon, including the crypts, mucus layers, lumen, and fluid flow. Using this model, we simulate gut environments including host cells with bacterial species alone and with a simplified gut microbiota (SIHUMIx). These simulations reveal patterns of host-microbe and microbe-microbe interactions that align with known findings. A key strength of the Virtual Colon is its ability to show how interactions unfold over time and space, offering new insights beyond traditional modeling approaches. The Virtual Colon is freely available and can be adapted to other structured biological environments (http://www.github.com/maringos/VirtualColon).

**KEYWORDS**  individual-based modeling, constraint-based modeling, host-microbiome interactions

**Peer Reviewer** Daniel Rios Garza, Evandro Chagas Institute, Ananindeua, Brazil

Address correspondence to Christoph Kaleta, c.kaleta@iem.uni-kiel.de.

Georgios Marinos and Johannes Zimmermann contributed equally to this article. They are listed alphabetically but have shared first authorship with an interchangeable order.

The authors declare no conflict of interest.

Host-microbial interactions are of crucial importance for host health and fitness (1). A well-studied example of localized host-bacterial interactions is the mammalian large intestine, where the gut microbiome provides the host with nutrients that otherwise would not be available to the host, protects it from colonization by pathogens, and interacts with its immune and nervous system (2). For instance, some colonic bacteria can ferment fibers and other components in the colonic lumen. Subsequently, they produce short-chain fatty acids (SCFAs), such as acetate, butyrate, and propionate, which have diverse beneficial properties for host health (3, 4). Acetate and propionate

are carried and utilized metabolically outside the intestine (5), while butyrate is used locally for energy generation (6). However, their effect on the host is not only metabolic, as there is evidence of an interaction between SCFA and the immune system (7).

However, the case of SCFA is far from exceptional (8). Therefore, it would be very informative to systematically explore host-microbial interactions inside the colonic microenvironment. In the colon, bacteria reside and are separated from the colonic tissue by two layers of a host-produced mixture of glycoprotein and oligosaccharides, collectively called mucus, which is located in between the lumen and the colonic epithelium (9). This environment is characterized by the limited presence of oxygen fostering the growth of anaerobic bacteria under physiological conditions (10, 11). The bacteria consume dietary compounds that are not absorbed in the small intestine along with host-derived compounds such as mucus and bile (12). Therefore, it is not surprising that the architecture of the colonic micro-environment offers a palette of well-studied localized host-bacterial interactions, as well as unknown ones.

Attempts to model this physicochemical environment and its phenomena in the intestine exist and are informative. For instance, Cremer and colleagues developed a mathematical, quantitative model that combines the microbes, their production of SCFA, and colonic water absorption (13). However, a promising method to understand novel, unexplored host-microbial interactions is the modeling of the metabolism of both host and bacterial cells. Recent progress in the field of Systems Biology has enabled the creation of models of metabolism for a variety of organisms. Specifically, starting from the genomic information of individual species (14, 15), it is possible to infer the possible metabolic pathways of the respective organisms and to organize them in networks of interconnected metabolic reactions, known as genome-scale metabolic models (16). As a further step, we can apply mathematical methods, such as linear programming, to simulate the input and output of these networks. A common mathematical approach is flux balance analysis (FBA), which optimizes the input and output of the models so that an objective function such as cell growth is maximized (17).

Simple metabolic models, such as the *Escherichia coli* core model with less than a hundred reactions, can be used to understand how the central metabolism of an organism operates (18). For more advanced research, one can utilize existing curated models (e.g., the extensive, human-associated bacterial models collection AGORA2) (19) or even create their own models tailored to the respective research question by employing modeling software such as gapseq and CarveMe (15, 20). Most importantly, it is possible to extend the layers of knowledge by combining data sets from different omics and databases, also beyond the strict boundaries of genome-scale metabolic modeling (e.g., see Virtual Metabolic Human database [21], MetaNetX [22–25], KEGG [26–28]). Emanating from this basis, spatiotemporal approaches that focus on simulating not only the models but also their environment through time have been developed (29). For instance, an advanced approach by Versluis and colleagues has simulated how the metabolism of specific gut bacteria could alter based on the changes of the environment cues (30).

In this study, we build on the metabolic modeling software BacArena (31), which is an agent-based framework for spatial and temporal simulations of metabolic models. Specifically, it allows the design of a chemically defined environment, where the cells acting as agents can have predefined properties (e.g., speed, chemotaxis, linear or exponential growth). Specifically, the models move, reside, duplicate, and die in a 2D space, where they can use the compounds around them. This setup allows the use of kinetic information, which is, however, difficult to obtain for nutritionally complex environments like the colon. To avoid unreasonably high uptake rates, the immediate surroundings of a microbial cell are modeled with appropriate dimensions that allow uptake in physiologically reasonable scales. In addition, parsimonious FBA is used to calculate the consumption and production of compounds as well as the production of biomass as an objective function over a defined time step. It provides the metabolic

profile of each and every model in the simulation operating under steady state at a given time, including the consumption and production of metabolites by each bacterium. The 2D space is then updated in terms of new concentrations of compounds and addition (duplication) or removal of models. These time steps can hence be modeled iteratively, giving a numerical solution to the dynamics of growth and metabolism of the modeled system (16, 31). We present a virtual colon-like 2D and context-specific application of BacArena that combines the host and bacterial sides into one single simulation framework. Our implementation spatially separates bacteria and host cells by representing multiple mucus layers. Furthermore, it enables a realistic simulation of host and microbial interactions, as it combines spatial-dependent diffusion phenomena that constrain substrate availability and mimic the lumen fecal stream powered by peristalsis activity. Lastly, BacArena can be found online (http://www.github.com/euba/BacArena), and it is required for running and adapting Virtual Colon. It is freely available on GitHub so that it can be easily adapted by future users (http://www.github.com/maringos/VirtualColon).

## MATERIALS AND METHODS

### General setup

We conducted computational colonization experiments mimicking an experimental setting with germ-free mice originally described by Geva-Zatorsky and colleagues (henceforward the reference study) (32). Therefore, each simulation comprised bacterial models together with host models in our colon-like in-silico setting. Furthermore, each setting simulates seven iterations of 1 h. The first iteration was used for the diffusion of compounds into the environment, to ensure that the compounds were well dispersed and readily available for uptake by the models. Subsequently, the simulation continued, combining metabolic models and diffusion. In total, each setting was replicated 10 times. The growth of each organism was calculated based on parsimonious FBA (33). The majority of colonization experiments refer to the introduction of one species (henceforward mono-colonization). There were also three cases of specific-pathogen-free mice. In that case, we employed the generic simplified human intestinal bacterial community of eight species, whose initial relative abundance in the simulation was calculated based on experimental data (see Table 1, SIHUMIx) (34).

### Metabolic models

In the case of the mono-colonization experiments, bacterial metabolic models matching the bacterial strains of the reference study were used (see Table S1) (32). If a genome of a strain was not available, we took the most similar genome from NCBI and literature (see Table S1) (37). The genomes were utilized to create the respective bacterial metabolic models by gapseq (version ce962ff) (15). The medium for gap filling was the one available in the lumen and mucus layers of the Virtual Colon (see Table S1). The SIHUMIx metabolic models were reconstructed using gapseq (version 0e7af55) (15) based on the published genomic information by Becker and colleagues (34). The gap-filling medium was predicted by gapseq. Finally, the models were adapted to grow on the medium available in the lumen and mucus layers of the Virtual Colon. Since we were interested in exploring the host-microbial interaction by finding the compounds that are exclusively of bacterial origin and taken up from the host cells, we had to exclude the possibility that the host cells use the dietary input of the bacterial models in parallel. Since host absorption primarily occurs in the small intestine and lumen content is therefore mainly available for microbial metabolism, we introduced a new "d" compartment into the bacterial models to account for an exclusive nutritional uptake of compounds from the lumen.

The host metabolic models originate from an adapted version of the published Recon 2.2 model (38). This generic model was expanded so that a new compartment

**TABLE 1** Overview of properties of the Virtual Colon

| Property | Information on values and units | Further comments | Reference |
|---|---|---|---|
| Grid cell dimension | $0.00025 \times 0.00025$ cm$^2$ | Default in BacArena | (31) |
| Y-axis size | 179 grid cells long | Set by the developers | Not applicable |
| X-axis size | 24 grid cells long | Set by the developers | Not applicable |
| Scaling factor for minimum weight and its standard deviation of host models | 132.73229:4.42 | A human COLO205 colon cell has a diameter of 13 µm; hence, its area is equal to 132.73229 µm$^2$. The area of a bacterium in BacArena is, by default, 4.42 µm$^2$. | (35) BNID 108890 |
| Scaling factor for maximum weight of host models | 1,000:70 | 70 cells represent the weight of around 1,000 cells of a human crypt | (35, 36) BNID 110648 |
| Starting amount of bacteria in the lumen | 29 | Set by the developers | Not applicable |
| Starting amount of bacteria in the outer mucus | 18 | Set by the developers | Not applicable |
| Starting amount of bacteria in the inner mucus | 9 | Set by the developers | Not applicable |
| Initial relative abundance of *Anaerostipes caccae* in SIHUMIx simulations | 0.1263867 | Based on the concentration in human feces [log10 cells/g dry matter]; see text | (34) |
| Initial relative abundance of *Bacteroides thetaiotaomicron* in SIHUMIx simulations | 0.1444797 | Based on the concentration in human feces [log10 cells/g dry matter]; see text | (34) |
| Initial relative abundance of *Bifidobacterium longum* in SIHUMIx simulations | 0.1388008 | Based on the concentration in human feces [log10 cells/g dry matter]; see text | (34) |
| Initial relative abundance of *Blautia producta* in SIHUMIx simulations | 0.1385367 | Based on the concentration in human feces [log10 cells/g dry matter]; see text | (34) |
| Initial relative abundance of *Clostridium butyricum* in SIHUMIx simulations | 0.1065769 | Based on the concentration in human feces [log10 cells/g dry matter]; see text | (34) |
| Initial relative abundance of *Clostridium ramosum* in SIHUMIx simulations | 0.1263867 | Based on the concentration in human feces [log10 cells/g dry matter]; see text | (34) |
| Initial relative abundance of *E. coli* in SIHUMIx simulations | 0.118859 | Based on the concentration in human feces [log10 cells/g dry matter]; see text | (34) |
| Initial relative abundance of *Lactobacillus plantarum* in SIHUMIx simulations | 0.09997359 | Based on the concentration in human feces [log10 cells/g dry matter]; see text | (34) |

of exchange reactions was available. This compartment contains exchange reactions ending with "u" and is a copy of the original compartment of exchange reactions labeled "e". The reason for this addition was to represent the flow of compounds between the bacteria in the lumen and the colonic cells (compartment "e") and between the colonic cells and the blood compartment (compartment "u"), following the strategy of Sahoo and Thelie (39). Moreover, we used iMAT to integrate mouse transcriptomic data from the reference study (32, 40) into the host models (41) to build context-specific models of the respective colonic tissue. After translating gene expression activity into reaction expression, a reaction was assumed to be lowly expressed if it was within the first quartile of all reaction expression values. A reaction was assumed to be highly expressed if it was in the fourth quartile across all reaction expression values. Moreover, we required a pre-set list of forced reactions and unconstrained nutrition-related exchange reactions. This information was used as an input to determine a context-specific model for each condition. All exchange reactions and the demand reaction for ATP "DM atp c" were available in the final models. It should be noted that although Recon 2.2 is a human model, it was highly curated and was much more tested than any specific mouse model at the time of simulations, while it requires less computational power than its newer versions. Furthermore, there is little difference in metabolic capacity between mice and humans (42), prompting us to use Recon 2.2 in our modeling. Detailed information on the utilized models and the procedures can be obtained from Table 1, and the models are available online (see Table S1 and folder "Supplementary Data S1" at www.doi.org/10.6084/m9.figshare.29282060). Both the host and bacterial models retained their original parameters unless otherwise specified.

## Representation of the colon09

The computational space of the Virtual Colon was developed by extending BacArena, which provides a two-dimensional grid area (31). One bacterial or host cell was located in one grid cell. The size of the virtual colonic lumen, the mucus layers, and the structure of the colonic crypts were calculated based on anatomic measurements converted to the dimensions of one grid cell. Specifically, thanks to a predefined crypt-like position of the host cells in the environment, two half-crypts were represented together with their common inter-cryptal space (see Fig. 1 and Table 1).

The Virtual Colon is composed of five spaces: lumen, outer mucus, inner mucus, intermediate space, and host space. The inner mucus's width was equal to one-third of the total mucus one, while the outer mucus's width was equal to two-thirds (43). The concentration of outer mucus layer compounds was found to be four times greater than that of the outer ones (44) and was used accordingly (see Table S2 for more information on design, calculation, and further references). The bacterial models were introduced into the colonic lumen and both mucus layers. The bacteria of the inner mucus were unable to move unless the inner mucus was partially degraded. Then, the bacterial models could move through the grid cell and potentially pass toward the host. The bacterial models could not enter human space, but they could be located next to the human models in the intermediate space.

We started the simulations with reported abundances from the reference study (32). The initial position of microbial cells in the Virtual Colon was random. Important additional parameters were the arena's size and volume; the actual total volume of the murine host's cecum, colon, and rectum ($0.37 \text{ cm}^3$) (45); and the bacterial input (200 µL of $10^8$ cfu/mL; D. L. Kasper, personal communication) (32). For instance, the amount of bacteria in the lumen was calculated based on the following formulas:

$$\text{Amount}_{\text{lumen}} = \frac{\text{area}_{\text{lumen}} \times \text{volume}_{\text{arena}} \times 0.2 \times 10^8}{\text{area}_{\text{total}}}$$

$$\text{Volume}_{\text{arena}} = \text{length}_{x\text{-axis}} \times \text{length}_{y\text{-axis}} \times \sqrt{\text{length}_{x\text{-axis}} \times \text{length}_{y\text{-axis}}}$$

Similarly, the amount of bacteria in the outer and inner mucus areas was calculated. In the case of the SIHUMIx community simulations, the same approach was followed. However, an abundance factor accounting for the relative abundances of each species was used (see Table 1) (34).

The virtual host cells inherited the default human class settings of BacArena (31). For the minimum weight and its standard deviation, the scaling factor was used to relate the area of human and bacterial cells. Rodriguez and colleagues have suggested that one crypt column is as long as 31.1 cells in the descending mouse colon (36). Similarly, the two-crypt setting provided 70 model slots for host cells in total. The maximum allowed weight of 70 cells represented a maximum total weight of around all cells of a human crypt (BNID 110648 [35]). In addition to these adjustments, no movement was allowed for the host models, and their growth was considered to be linear through time (see Table 1). As noted previously, the host models have the demand reaction for ATP "DM atp c"; therefore, it would have been possible to optimize the ATP maintenance instead of biomass. However, it should be noted that while there is some debate about whether optimization of growth is an adequate objective function for flux balance simulations of colonic cells, the biomass function includes all molecular components of cells, which are also required for cell renewal. Furthermore, there is constant growth and shedding of colonic cells with an estimated turnover time of 3–5 days in mice (46). Thus, in the colonic context, optimization of growth is a valid objective function to predict metabolic fluxes.

The transport of molecules inside BacArena (31) was modeled by diffusion (see Table S2). As discussed above, the arena was split into the following diffusion sectors: lumen, outer mucus layer, inner mucus layer, intermediate layer in-between, and blood. The utilized diffusion coefficients were calculated using a power regression function

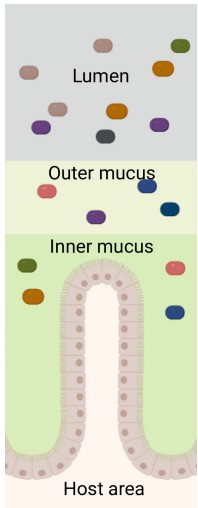

**FIG 1** Schematic representation of the virtual colonic space. The colored rectangles represent the models of the microbial species, and the two shades of green represent the two mucus layers above the host's epithelium. Created in BioRender (C. Kaleta, 2025. https://BioRender.com/2pwtd0t).

(coefficients vs molecular weight) for the colonic lumen and mucus layers. The molecular weight was used for nutritional compounds. For other compounds, the default value of glucose was assumed. The regression functions were generated using the intestinal diffusion data of Winne and colleagues (47). The same strategy was followed for the restricted areas and the lumen (48). Especially for mucus layers, the diffusion coefficients of molecules in outer mucus were equal to two-thirds of mucus diffusion coefficients, while for inner mucus, they were equal to one-third of mucus diffusion coefficients, based on the observation that one-third of the length of the mucus belongs to the inner layer (43). Furthermore, the mucus compounds were not allowed to diffuse to represent the stability of the mucus in the colonic environment. Oxygen could only diffuse to host areas to ensure limited "oxygenation" of the Virtual Colon. Practically, the diffusion of compounds in the lumen was designed to be less restricted than in mucus layers, while not possible in the intermediate layer and in blood. At the same time, diffusion of blood molecules was possible in the intermediate layer and in the blood.

## Nutrition

Following the nutrition that was chosen by the authors of the reference publication (32), the primary source of nutritional compounds was the LabDiet JL Rat and Mouse/Auto 6F 5K67. These compounds were matched to the dietary bacterial exchange reactions to exclude nutritional competition between host and bacterial cells. The intestinal absorption and the dimensions of the Virtual Colon were considered for calculating the exact nutritional input. It is worth noting at this point that BacArena—and therefore Virtual Colon—is a simulation software that assumes that the concentration of the compounds can be used as the lower bounds of the respective exchange reactions. In addition to the LabDiet, some components such as water, $H^+$, minerals, vitamins, and mucus compounds were added as part of the lumen compartment. To represent the inflow of nutrients from the blood circulation, a minimal collection of compounds was added into the blood compartment. Their concentration was based on the literature (e.g., BNID 110365 [35]) and various Human Metabolome Database entries (https://hmdb.ca/ [49–52]). To infer these components, a shadow cost analysis based on the BacArena (31) function "plotShadowCost" was utilized (see folder "Supplementary Data S2" at www.doi.org/10.6084/m9.figshare.29282060). A list of compounds can be found in Table S2.

## Software

The simulations and analyses thereafter presented in this manuscript were conducted in the R environment (version 4.2.1) (53). Optimizations were generally conducted using the BacArena application (version 1.8.2, commit euba/BacArena@dc8e591 and euba/BacArena@008d186) (31) and sybil (version 2.2.1, commit SysBioChalmers/sybil@58eb989) (54). The linear programming solver was academic version 22.1.0 (55), which is available through the R package cplexAPI (version 1.4.0) (56). For data management, the software tidyverse (version 2.0.0) (57) and especially its library dplyr (version 1.1.4) (58) were used. For plotting, the software ggplot2 (version 3.5.1) (59), ggvenn (version 0.1.10) (60), ggthemes (version 5.1.0) (61), scales (version 1.3.0) (62), ggalluvial (version 0.12.5) (63), and BacArena (version 1.8.2, commit euba/BacArena@dc8e591) (31) were utilized. The software devtools (version 2.4.5) (64) was also used to manage the R environment.

The simulations were executed in the high-performance computing center available at Kiel University as an array of independent tasks, each on a single CPU core. The longest running time was observed for the SIHUMI simulations, which included 10 replicates running in a for-loop. A total runtime of 4.5 h and a maximum memory usage of 6 GB were recorded. Although these data heavily depend on the randomly assigned node in the cluster, they are indicative of the performance and scalability of the setting. The underlying BacArena framework scales linearly with the number of individuals added (see also Fig. 2 in the BacArena manuscript [31]).

## RESULTS

The main objective of this study was to provide a computational framework to simulate host-microbial interactions in the mammalian colon. We first used our approach to study host-microbiome interactions in the colon based on a previously published data set of single-strain colonization experiments in germ-free mice (32). Besides microbial abundance data, Geva-Zatorsky and colleagues provided extensive transcriptomic data sets of these colonization experiments. Based on those transcriptomes, we reconstructed context-specific models of colonic tissue for the respective mice. For our study, we used context-specific Recon 2.2 (38) host models along with bacterial models for joint simulations in a virtual colon.

The Virtual Colon provides detailed information on the host-microbial metabolic interactions in space and time. To reveal those interactions in the mono-colonization experiments, we extracted all fluxes across all time points for each compound and for each model. Since a positive flux shows production and a negative one shows consumption of a compound, we calculated the sum of fluxes of all models for each compound. Collectively, the host models produced 20 compounds, and the bacterial ones produced 27 compounds (Fig. 2A). Interestingly, the host-microbial interactions seem to be very pair specific since we could identify compounds that in some pairs are bacterially produced or in other pairs by the host (e.g., glycine, L-alanine; Fig. 2A). Furthermore, we observed the host exchange of nitrogen-containing compounds, toward the bacterial models, and the uptake of bacterially produced SCFA. Additionally, we found that although the majority of the bacteria interacted metabolically with the host regardless of their position (e.g., see cases with fluxes of models originated from all three layers: lumen, outer mucus, inner mucus; Fig. 2B), we also observed cases of localized interactions (Fig. 2B).

Further, we employed the Virtual Colon to study host-bacterial interactions with the SIHUMIx bacterial community. As a proof-of-principle, we studied the growth patterns of the bacterial and host models. Each bacterial strain achieved different growth rates, and all the models were still in the exponential phase at the end of simulation. The host cells did not reproduce, as expected (Fig. 3A). Regarding the different layers of Virtual Colon, there is a variation of growth rate of the bacterial models depending on their position in the colonic space (Fig. 3B). Inside the inner mucus layer, they could only marginally spread in comparison to the other locations due to the design of the movement barrier.

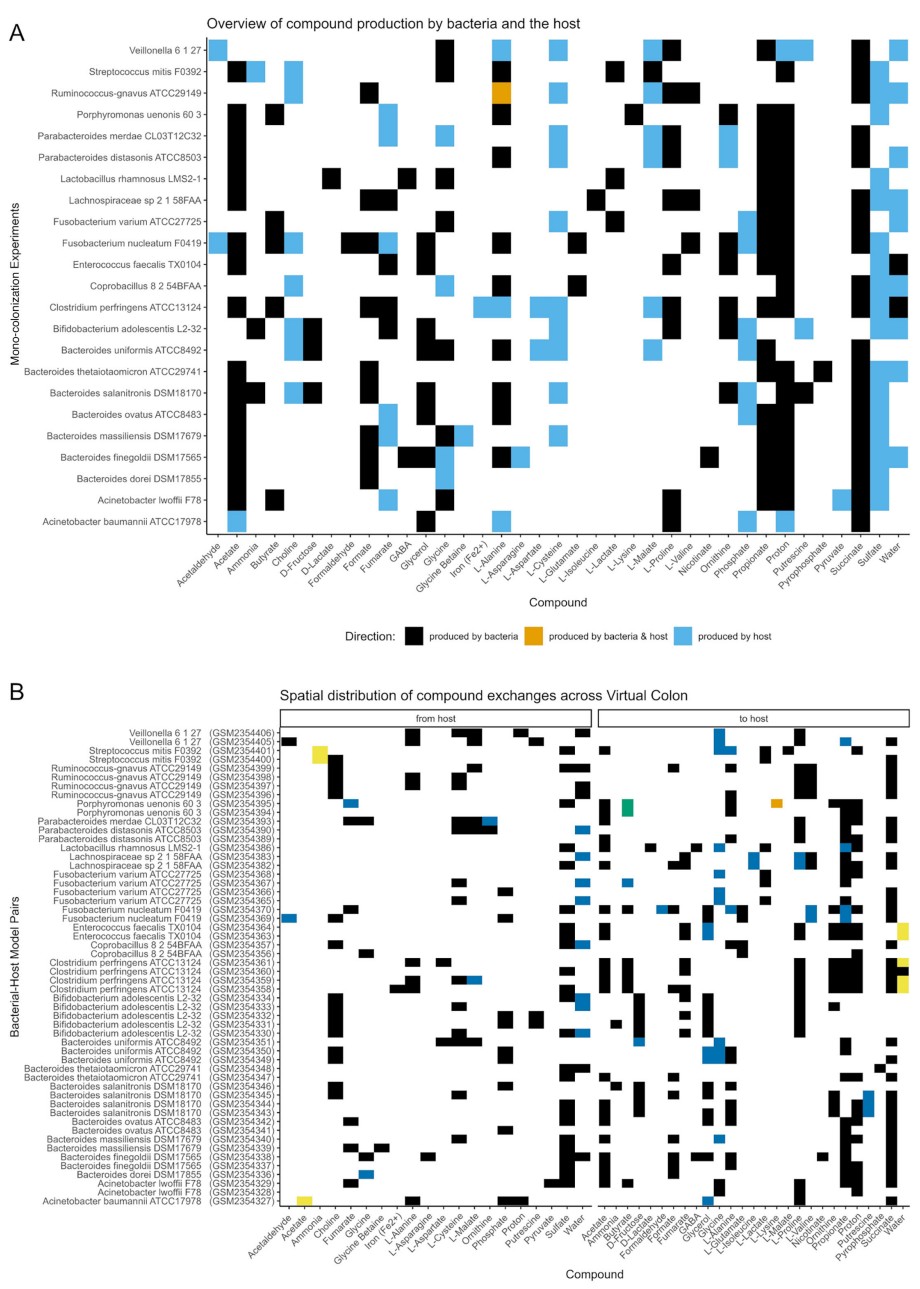

FIG 2  (A) Exchange of compounds which are produced by the bacterial models or context-specific host models. The color of the layers depends on the origin of the producer. Multiple context-specific host models, each originating from a transcriptomic data set of mono-colonization experiments, were simulated per bacterial model. (B) Spatial localization of exchanges: exchange of compounds which are produced by the host models and are consumed by the bacterial model, and vice versa. Each row represents an independent set of simulations of a specific bacterial species with a distinct context-specific host metabolic model. The repetition of bacterial strain names (e.g., *Streptococcus mitis*) highlights that the same bacterial species was simulated with multiple context-specific host models, identified by their respective codes of the transcriptomic data (e.g., GSM235440 and GSM2354401). To estimate the exchange of compounds among the models, the sum of all fluxes across time for each compound and for each model was determined. A positive sum denotes production, while a negative one denotes consumption. To limit the numerical artifacts, the fluxes whose absolute value was less than 1E−6 were considered zero.

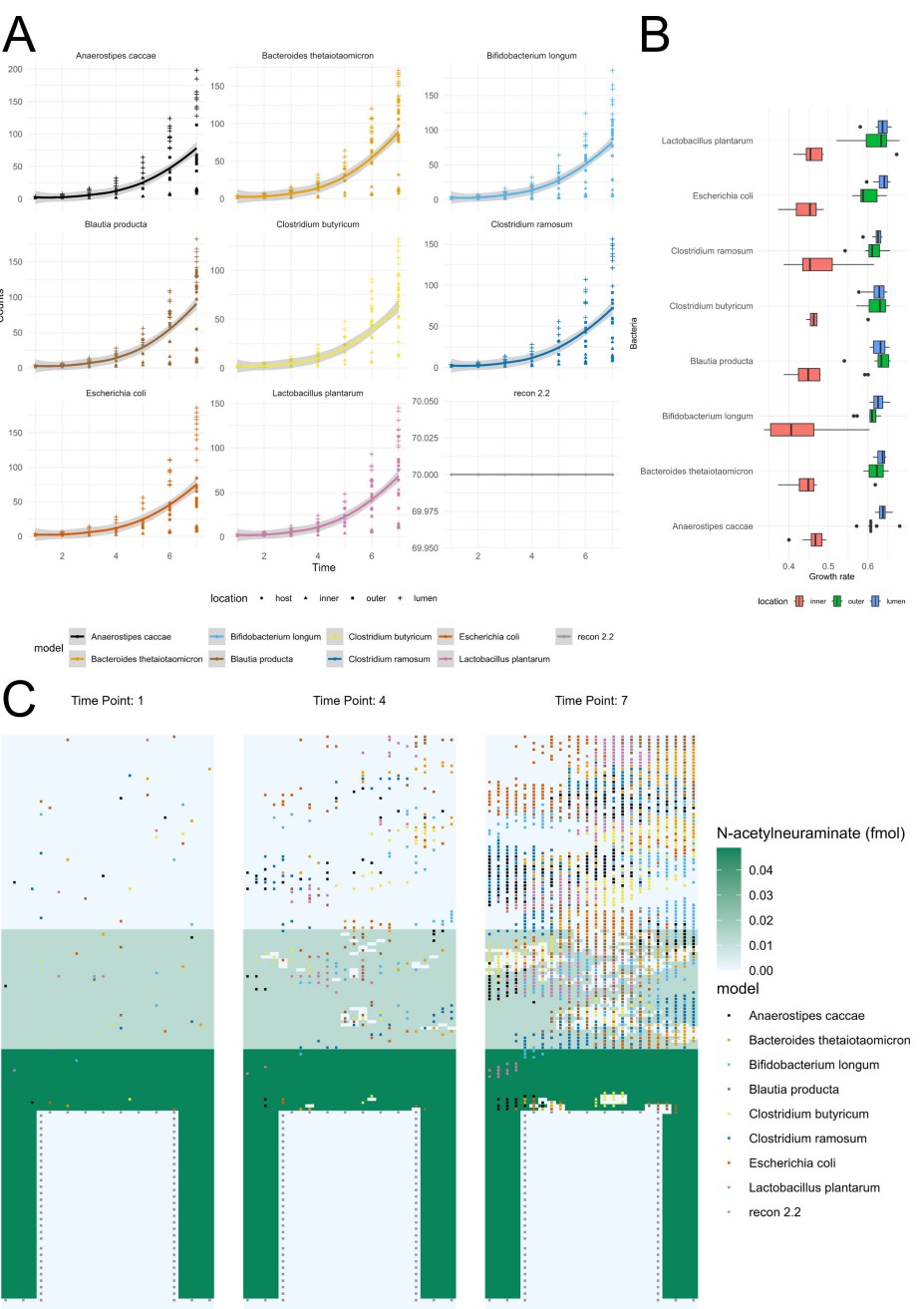

**FIG 3** (A) Growth curves for the individual bacterial and host models. The data points originate from all the replicates of the GSM2354402 SIHUMIx simulation. The individual lines are plotted using the "loess" smoothing function. (B) Variation in growth rates (unit h⁻¹) by location in Virtual Colon and species. The growth rate data points represent all replicates from the GSM2354402 SIHUMIx simulation. (C) Schematic representation of the mucus compound N-acetylneuraminate through three time points in a SIHUMIx-host simulation (GSM2354402, 10th replicate). Each subplot represents the substance concentration and bacterial population per time point in hours.

The SIHUMIx microbial community that resides in inner and outer colonic mucus also gradually consumed the mucus compound N-acetylneuraminate. Most interestingly, the decreasing quantity of the compound follows the growth of the bacteria models in the area. The bacterial models of *B. thetaiotaomicron*, *B. producta*, and *C. butyricum* can degrade N-acetylneuraminate (Fig. 3C).

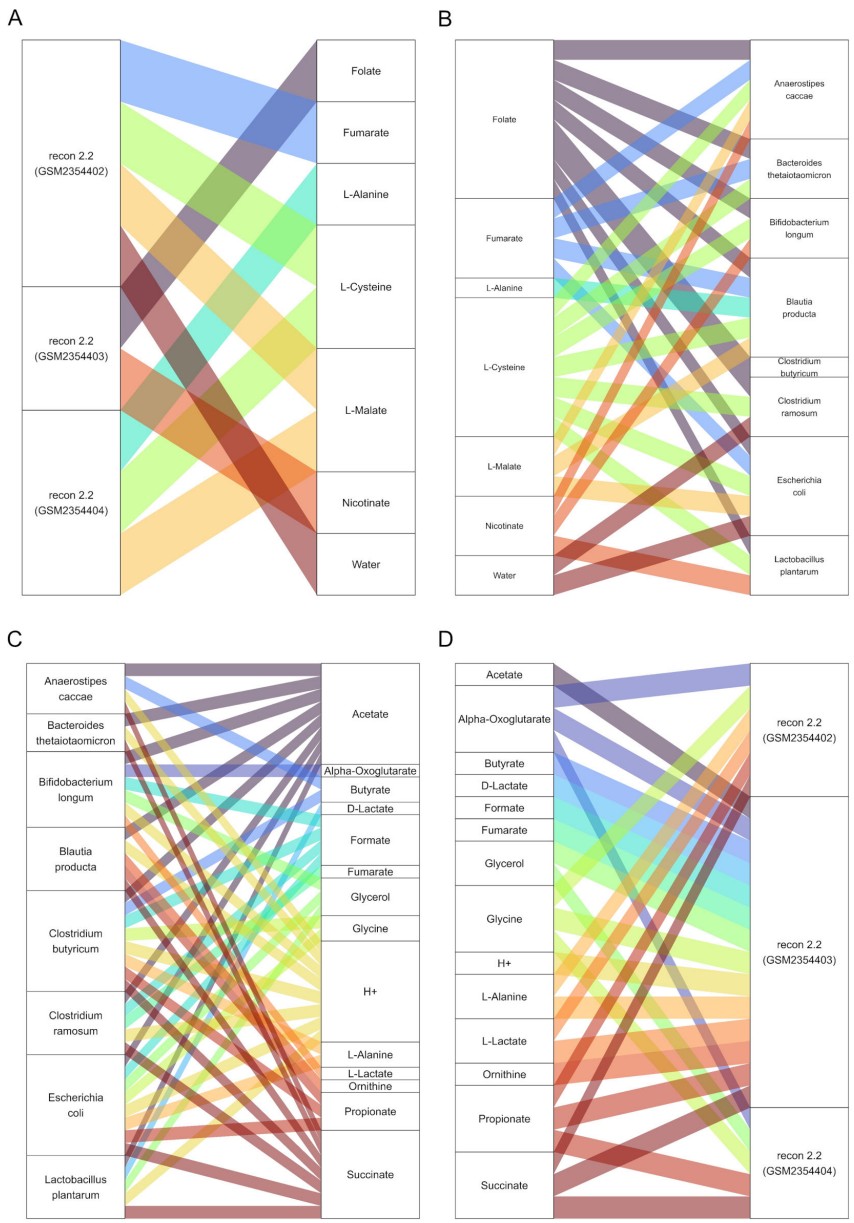

**FIG 4** Schematic representation of compounds produced by the context-specific host models (A) and consumed by the SIHUMIx bacteria (B) along with the bacterially produced ones (C) that are consumed by the host models (D). Metabolites with the highest interaction frequency (≥50%) across all time points and simulations were selected to be plotted for simplicity. For the full data set, see Table S3.

The observation of microbial mucus degradation led to more general questions on what compounds are consumed or produced by the metabolic models. The host models, similarly to the mono-colonization experiments, are context-specific Recon 2.2 models (38), whose reconstruction was based on the transcriptomes GSM2354402, GSM2354403, and GSM2354404, respectively. While those three models produce and consume mostly the same compounds, they do have distinct metabolic profiles that can be attributed to the different transcriptomes. For instance, the recon model GSM2354403 produced L-alanine, which was taken up by the bacteria, while the recon model GSM2354404 consumed bacterial L-alanine (Fig. 4A and D). However, all models consumed bacterially produced SCFAs, such as formate, acetate, propionate, and butyrate (Fig. 4C and D).

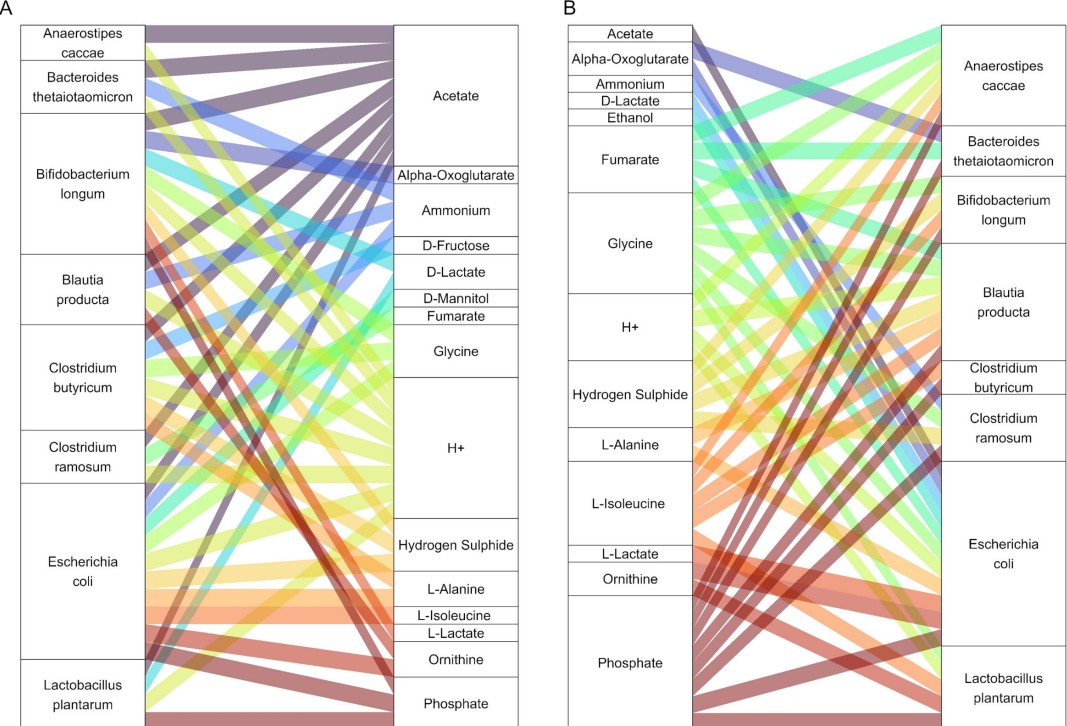

**FIG 5** Schematic representation of compounds produced by the SIHUMIx bacterial models (A) and consumed by other SIHUMIx bacterial models (B). Metabolites with the highest interaction frequency (≥50%) across all time points and simulations were selected to be plotted for simplicity. For the full data set, see Table S3.

Interestingly, the models exchange various metabolic compounds (e.g., water, vitamins, amino acids, TCA-cycle compounds such as alpha-oxoglutarate).

Not only were the bacterially produced compounds taken up by the host models, but they were also consumed by other bacterial species of the community (Fig. 5A and B). For instance, apart from amino acids and carbon sources, they also exchanged metabolites like protons, ammonium, phosphate, and hydrogen sulfide. It is possible to characterize the metabolic role of community members. The bacteria model of *E. coli* was the most active species in terms of production and consumption of compounds. On the other hand, *B. longum* and *C. butyricum* contributed more compounds than they consumed, while the opposite was true for *A. caccae* and *B. producta*.

Furthermore, in order to understand how the metabolic interactions change when comparing the interactions from mono-colonization with the SIHUMIx simulations, we investigated which compounds are unique in each of the two settings. It can be observed that the exchanged compounds were mostly common (Fig. 6). However, the majority of the unique compounds in mono-colonization experiments were nitrogen-containing compounds. In the other two parts of the Venn diagram, we could identify amino acids, metabolism intermediates (e.g., fumarate), and SCFAs.

## DISCUSSION

Host-bacterial metabolic interactions are considered important drivers of host health and fitness. Therefore, one of the objectives of this study was to develop a data-driven approach that allows us to systematically explore the compounds that are exchanged between the host and the bacterial cells. Here, we present the Virtual Colon that extends the agent-based modeling framework BacArena (16, 31) to study host-microbiome interactions in the mammalian large intestine. Specifically, it enables host-microbial simulations, in which bacteria and host cells are spatially separated through multilayer mucus layers. Moreover, it includes spatial-dependent diffusion that constrains substrate

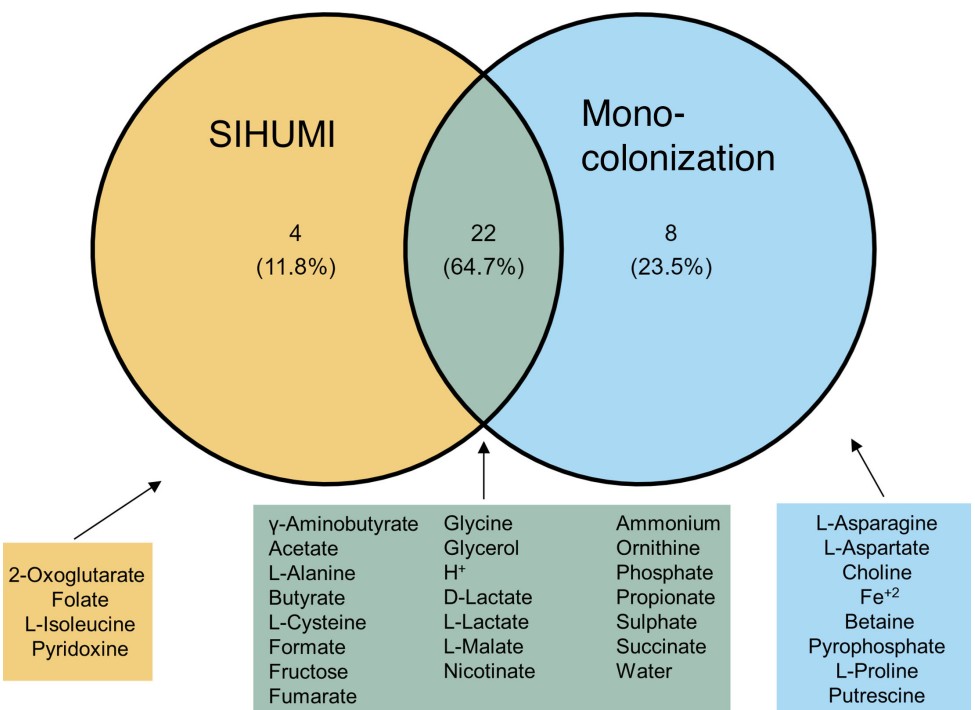

**FIG 6** Schematic representation of compounds exchanged by the bacterial models and host models in the mono-coloniza-
tion and SIHUMIx experiments. Only interactions with bacteria from the shared genera were taken into account (*Bacteroides,*
*Bifidobacterium, Clostridium, Lactobacillus*). All simulation time points were included in the analysis.

availability for the models. Most importantly, it not only enables the joint simulation of
host and microbial interactions but also visualizes them. While other methods have
contributed to the field either by using differential equations and/or agent-based
modeling (Table 2), the Virtual Colon, which is built on BacArena, combines them in one
software framework.

In our simulations, we utilized host models that originated from the human model
Recon 2.2 (38). For the simulations, the host models were adapted using transcriptomic
data (40) from mono-colonization experiments of germ-free mice with the respective
species (32). In the case of colonization with non-specific pathogens, we used the
SIHUMIx bacterial models that comprise a generic eight-species human intestinal
bacterial community (34).

To continue with the host-bacterial interactions in the mono-colonization experi-
ments, the observed usage of cysteine by some bacteria of the genus *Fusobacterium*
and *Clostridium* is in line with the literature, which proposes that cysteine is metabolized
by bacteria, leading to hydrogen sulfide (69). Hydrogen sulfide has been characterized
as a compound with multiple effects, as in low levels, it supports colonic health, while
in excess, it is associated with inflammatory and pathological states (e.g., inflammatory
bowel disease and cancer) (69).

Additionally, the compound 4-aminobutanoate, also known as γ-aminobutyric acid
(GABA), is predicted to be produced by *Lactobacillus rhamnosus* and *Bacteroides finegoldii*
and be consumed by the host models, which is in accordance with the literature (70).
This compound may act on the central nervous system as well as the enteric nervous
system of the host as part of the microbiota-gut-brain axis (70). On the other hand,
*B. finegoldi* has not been reported to be a GABA producer. However, a recent article
proposed that the Bacteroides species can produce this molecule, although, in the
simulations, no other *Bacteroides* species did so (71).

Recently, a study based on metabolomics focusing on the colonic absorption of
compounds with low molecular weight in mice identified some of the computationally

**TABLE 2** Comparison of Virtual Colon with other published approaches[a]

| Name | Approach | Microbes | Scope | Host | Space | Time | Mucus | Gut motility | Availability | Ref |
|---|---|---|---|---|---|---|---|---|---|---|
| Gut-Microbiota | PDE model with fluid dynamics (Stokes) | Movement + predefined functions | Colon | Nutrient uptake | ✓ | ✓ | ✓ | ✓ | Matlab, https://forgemia.inra.fr/simon.labarthe/gut-microbiota | (65) |
| MicroGutPop | ODE model (Monod equation) | Predefined functional groups | Colon | – | – | ✓ | ✓ | ✓ | R, https://github.com/HelenKettle/microPopGut-Code | (66) |
| GutLogo | ABM | | Ileum | – | ✓ | ✓ | ✓ | ✓ | https://github.com/GutLogo/GutLogo | (67) |
| Synthetic Gut Microbiome | ODE model (gLV) | 25 microbes | Butyrate production | – | – | ✓ | – | – | Julia, https://github.com/RyanLincolnClark/Design-SyntheticGutMicrobiomeAssemblyFunction | (68) |
| Virtual Colon | ABM + PDE | Movement + metabolic model | Colon | Metabolic model | ✓ | ✓ | ✓ | ✓ | R, http://www.github.com/maringos/VirtualColon | This study |

[a]PDE: partial differential equation; ODE: ordinary differential equations; gLV: generalized Lotka-Volterra; ABM: agent-based modeling; Ref: reference; ✓, feature present; –, feature absent.

predicted compounds, which were subject to metabolic exchange in our simulations: (glycine) betaine, choline, alanine, glutamine, glycine, isoleucine, lactate, ornithine, and SCFAs (Fig. 2A and B) (72). Interestingly, host-derived glycine betaine and choline can be further bacterially metabolized to trimethylamine-N-oxide and trimethylamine, for which a linkage to atherosclerotic cardiovascular disease has been demonstrated (73).

While mono-colonization experiments offered a detailed overview of pairwise host-bacterial interactions, the bacterial species found in the mammalian colon are quite diverse. Therefore, we selected the representative SIHUMIx community to investigate how our results change in the context of a more complex community. Our simulations showed a wide exchange of compounds among the species as well as between bacteria and host cells. Not surprisingly, many byproducts were found also in mono-colonizations, implying that the repertoire of metabolic exchanges may be genera-dependent (Fig. 6). The bacterial models create a complex web of compounds, where carbohydrate and amino acid metabolism as well as cofactors (e.g., vitamins) and byproducts are the most common end products. This observation becomes intriguing if one considers that carbohydrates rather than amino acids are primarily used in bacterial energy metabolism, as amino acid metabolism is less cost-effective (74). Moreover, this finding suggests that mono-colonization experiments only partially capture interactions that can be observed in community colonization, as the metabolic repertoire and signature may alter significantly.

It must be stated that the predicted exchange of compounds between the host and the bacterial models is based on the assumption that the host colonic cells can exchange compounds from the luminal side of the colon. For instance, amino acids are known to be absorbed in the small intestine, while their absorption fate in the large intestine is unclear. While the colonic bacteria can metabolize the remaining amino acids, it is proposed that some of them can be used by colonocytes (75). In parallel, it has been shown that colonocytes can excrete amino acids (e.g., alanine, aspartate, asparagine, isoleucine, and valine) in both apical and basolateral directions *in vitro* (76). Therefore, the observed production or consumption of several compounds by the host is feasible and waits for further experimental validation. These experiments should investigate the transport of metabolites across the intestinal barrier. Such a pivotal set of experiments was performed by Li and colleagues, who identified gut species that deplete and metabolize amino acids throughout the intestine and therefore regulate host metabolism (77).

Furthermore, the Virtual Colon includes spatial parameters tailored to the large intestinal environment. For instance, the diffusion of compounds inside the mucus layer is limited, and the bacteria in the inner mucus cannot move unless mucus is consumed by community members, such as *B. thetaiotaomicron*, a known mucus-degrading species (78) (Fig. 3). Additionally, our results proposed that there are spatial, localized interactions; therefore, it could be suggested that the mucus layers can affect bacterial

metabolism (Fig. 2B). In line with those observations, the growth rates across all bacteria in the outer mucus layer seem to be the ones in the lumen, while the rates in the inner mucus layer are the slowest (Fig. 3B). Therefore, the simulation data suggest that our mucus resembles the function of the mucus both as a physical barrier and, to some extent, as a nutrient pool for the bacteria (9).

While the setting of the Virtual Colon recapitulates some of the observed host-microbial functions, there is also room for further expansions in the simulation setting. Future computational experiments may include more complex bacterial communities along with an improved representation of the abundance of bacteria with respect to the size of the simulated colon (e.g., the ratio of microbial abundance over the number of host models). Currently, the virtual gut represents the host model as the same size as bacterial models due to the technical limitations of BacArena. Another improvement could include metabolomic data from colonic tissue and lumen so that the chemical environment is as realistic as possible. Additionally, single-cell transcriptomics would allow the representation of the various cell types of the host tissue. To improve the realism of the simulations, the Virtual Colon also has the capacity to mimic the lumen feces stream powered by peristalsis activities, to mimic the replenishment of mucus compounds through time, or to initiate simulation assuming a non-empty colonic environment. Also, it would be interesting to investigate whether the variability observed in our simulations recapitulates inter-individual variances in microbiome composition in mice.

In summary, our novel simulation framework, Virtual Colon, builds upon the popular BacArena platform to create a comprehensive tool for studying host-microbe interactions in the colon. By spatially separating bacteria and host cells and incorporating multilayer mucus representation, the Virtual Colon enables the realistic simulation of host-microbial interactions. This is achieved through a spatial-dependent diffusion mechanism that constrains substrate availability and accurately mimics the fecal stream. Furthermore, the Virtual Colon allows for the joint simulation of host and microbial interactions, which we demonstrated for human gut microbiome communities and personalized host models using transcriptomic data. Thus, Virtual Colon allows for the personalized modeling of host-microbiome interactions given that host transcriptomic and compositional microbiome information is available. Notably, we employed the Virtual Colon to identify species-specific mucus layer degradation and amino acid exchange between microbes and host, providing valuable insights into the intricate dynamics of the gut ecosystem. Our freely available software on GitHub invites future researchers to adapt and expand its capabilities to advance host-microbe interaction studies.

## ACKNOWLEDGMENTS

C.K. acknowledges support from the German Research Foundation for the support through the collaborative research center 1182 Origin and Function of Metaorganisms (sub-project A1), the research unit miTarget (FOR5042), the excellence cluster "Precision medicine in chronic inflammation" (EXC2167), and additionally the German Ministry for Education and Research (E:Med iTREAT, support code 01ZX1902A).

This research was supported, in part, through high-performance computing resources available at the Kiel University Computing Center. The authors thank Dr. Ahmed Samer Kadib Alban for providing critical opinions on the manuscript.

Artificial intelligence was utilized in improving the readability and clarity of the article, which was reviewed and approved by the coauthors.

Conceptualization: G.M., J.Z. Methodology: G.M., J.Z., J.T., C.K. Software: G.M., J.Z. Formal analysis: G.M. Investigation: G.M., J.Z., J.T. Resources: C.K. Data Curation: G.M. Writing—original draft: G.M., J.Z. Visualization: G.M. Supervision: C.K., J.Z. Project administration: G.M. Funding acquisition: C.K. Writing—review and editing: all coauthors.

## AUTHOR AFFILIATIONS

[1]Research Group Medical Systems Biology, University Hospital Schleswig-Holstein Campus Kiel, Kiel University, Kiel, Schleswig-Holstein, Germany

[2]CAU Innovation GmbH, Kiel University, Kiel, Schleswig-Holstein, Germany

[3]Antibiotic Resistance Group, Max Planck Institute for Evolutionary Biology, Ploen, Schleswig-Holstein, Germany

[4]Evolutionary Ecology and Genetics , Zoological Institute, Kiel University, Kiel, Schleswig-Holstein, Germany

[5]Junior Research Group Mechanisms of Microbial Metabolic Interactions, Friedrich Schiller University Jena, Jena, Thüringen, Germany

## AUTHOR ORCIDs

Georgios Marinos  http://orcid.org/0000-0002-6443-7696
Johannes Zimmermann  https://orcid.org/0000-0002-5041-1954
Jan Taubenheim  https://orcid.org/0000-0001-7283-1768
Christoph Kaleta  http://orcid.org/0000-0001-8004-9514

## AUTHOR CONTRIBUTIONS

Georgios Marinos, Conceptualization, Data curation, Formal analysis, Investigation, Methodology, Software, Validation, Visualization, Writing – original draft, Writing – review and editing | Johannes Zimmermann, Conceptualization, Data curation, Supervision, Writing – review and editing | Jan Taubenheim, Conceptualization, Supervision, Writing – review and editing | Christoph Kaleta, Conceptualization, Supervision, Writing – review and editing

## DATA AVAILABILITY

The software is freely available on GitHub (www.github.com/maringos/VirtualColon, commit: 0e56bda). The supporting files, as well as the mono-colonization simulation files (folder "Supplementary Data S3") and the SIHUMIx simulation files (folder "Supplementary Data S4"), which are related to this paper can be found on Figshare (www.doi.org/10.6084/m9.figshare.29282060).

## ADDITIONAL FILES

The following material is available online.

### Supplemental Material

**Table S1 (mSystems01391-25-s0001.xlsx).** Sequence data for each species.
**Table S2 (mSystems01391-25-s0002.xlsx).** Design, calculation, and further references regarding mucus compounds.
**Table S3 (mSystems01391-25-s0003.xlsx).** Predicted interactions in SIHUMIx.

### Open Peer Review

**PEER REVIEW HISTORY (review-history.pdf).** An accounting of the reviewer comments and feedback of the first submission as well as our responses

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
