## [Reviewer comments · mSystems]

Virtual Colon: Spatiotemporal modelling of metabolic interactions in a computational colonic environment

Georgios Marinos, Johannes Zimmermann, Jan Taubenheim, and Christoph Kaleta

Corresponding Author(s): Christoph Kaleta, Christian-Albrechts-Universitat zu Kiel

Review Timeline:

Submission Date:

October 9, 2025

Accepted:

November 5, 2025

Editor: Babak Momeni

Reviewer(s): Disclosure of reviewer identity is with reference to reviewer comments included in decision letter(s). The following individuals involved in review of your submission have agreed to reveal their identity: Daniel Rios Garza (Reviewer #1)

Transaction Report:

DOI: <https://doi.org/10.1128/msystems.01391-25>

Re: mSystems01391-25 (Virtual Colon: Spatiotemporal modelling of metabolic interactions in a computational colonic environment)

Dear Prof. Christoph Kaleta:

Your manuscript has been accepted, and I am forwarding it to the ASM production staff for publication. Your paper will first be checked to make sure all elements meet the technical requirements. ASM staff will contact you if anything needs to be revised before copyediting and production can begin. Otherwise, you will be notified when your proofs are ready to be viewed.

Sincerely,
Babak Momeni
Editor
mSystems

Reviewer #1 (Comments for the Author):

Thank you for reviewing and correcting the manuscript. All my comments were appropriately addressed.

Reviewer #2 (Comments for the Author):

I am pleased that you addressed all of my comments and concerns. Best of luck!